# Surface Electromyographic Analysis of the Suprahyoid Muscles in Infants Based on Lingual Frenulum Attachment during Breastfeeding

**DOI:** 10.3390/ijerph17030859

**Published:** 2020-01-30

**Authors:** Ellia Christinne Lima França, Lucas Carvalho Aragão Albuquerque, Roberta Lopes de Castro Martinelli, Ilda Machado Fiuza Gonçalves, Cejana Baiocchi Souza, Maria Alves Barbosa

**Affiliations:** 1Medical School, Graduate Programme in Health Sciences, Federal University of Goiás, Goiânia, Goiás 38177, Brazil; maria.malves@gmail.com; 2Department of Neuropsychiatry, Federal University of Pernambuco, Recife, Pernambuco 38318, Brazil; fono_lucas@hotmail.com; 3Dental School, University of São Paulo, Bauru, São Paulo 223581, Brazil; robertalcm@gmail.com; 4Dental School, Federal University of Goiás, Goiânia, Goiás 38177, Brazil; ildafiuza@yahoo.com.br; 5Course Phonoaudiology, Pontifical Catholic University of Goiás, Goiânia, Goiás 38177, Brazil; cejana_f@hotmail.com

**Keywords:** lingual frenulum, tongue tie, electromyography, suprahyoid muscles, breastfeeding

## Abstract

Muscle electrical activity analysis can aid in the identification of oral motor dysfunctions, such as those resulting from an altered lingual frenulum, which consequently impairs feeding. Here, we aim to analyze the suprahyoid muscle electrical activity of infants via surface electromyography, based on lingual frenulum attachment to the sublingual aspect of the tongue and floor of the mouth during breastfeeding. In the present study, we have studied full-term infants of both genders, aged between 1 and 4 months old. The mean muscle activities were recorded in microvolts and converted into percent values of the reference value. Associations between the root mean square and independent variables were tested by one-way analysis of variance and Student’s t-test, with a significance level of 5% and test power of 95%, respectively. We evaluated 235 infants. Lower mean muscle electrical activity was observed with the lingual frenulum attached to apex/lower alveolar ridge, followed by attachment to the middle third/lower alveolar ridge, and between the middle third and apex/lower alveolar ridge. Greater suprahyoid muscle activity was observed with lingual frenulum attachment to the middle third of the tongue/sublingual caruncles, showing a coordination between swallowing, sucking, and breathing. Surface electromyography is effective in diagnosing lingual frenulum alterations, the attachment points of which raises doubt concerning the restriction of tongue mobility. Thus, it is possible to identify oral motor dysfunctions.

## 1. Introduction

Breastfeeding is considered essential for the promotion and protection of children’s health, due to the nutritional and immunological properties of breast milk, which protect children from respiratory diseases, infectious diseases, [1,2,3], and diarrhea [1,2,3,4]. In addition, breastfeeding is considered important for the adequate development of the stomatognathic system, because the removal of breast milk involves intense muscle activity [5,6].

During sucking in the womb, the suprahyoid muscles (digastric, mylohyoid, geniohyoid, and stylohyoid) effectively participate in the movement and stabilization of the mandible and tongue movement [7].

The correct movement of the tongue during breastfeeding promotes an adequate fit between the infant’s mouth and the mother’s nipple, compressing the nipple against the hard palate and favoring the removal of the milk due to the vacuum created in the oral cavity by the raising and lowering movement of the tongue [8,9,10,11]. 

The lingual frenulum is located in the inferior surface of the tongue and is considered a median fold of tunica mucosa that connects the tongue to the floor of the mouth and allows the free movement of its anterior part [12].

After apoptosis, the remaining residual embryonic tissue (lingual frenulum) may limit tongue movements to varying degrees. This congenital oral anomaly is referred to as ankyloglossia [9] and may cause reduced mouth opening, imprecision, and restriction of isolated tongue movements, a heart-shaped tip, downward protrusion [13], the tongue to rest on the floor of the mouth, and difficulties in sucking, chewing, swallowing, and speech functions, among others [12,13,14,15].

Oral dysfunctions caused by an altered lingual frenulum (tongue tie) may compromise breastfeeding by causing discomfort and pain to mothers, ineffective emptying of the breast, poor weight gain, and/or early weaning [12].

Studies that have evaluated the lingual frenulum by quantitative and objective methods are scarce and have tried to determine whether the anatomical findings could compromise tongue movement, and consequently oral functions [8]. Such methods enable safe diagnoses to guide therapeutical procedures for the continuity of breastfeeding.

Research on the use of surface electromyography (EMG), which is a method for recording variations in muscle electrical activity during contraction [16], is also scarce, especially in terms of the evaluation of sucking function in infants [7,17,18,19,20,21,22,23,24,25,26]. EMG is considered an easy, fast, low-cost, safe, and non-invasive procedure that can provide important information about muscle function [7,16,17,23,24,25,26], which can be used to diagnose oral motor dysfunction accurately [23,24]. 

Some studies have evaluated the activity of orofacial muscles using surface electromyography during breastfeeding and other feeding methods [7,18,19,20,21,22,23,24,26]. However, to the best of our knowledge, no study has analyzed the activity of the suprahyoid muscles in newborns and infants in terms of the sucking function and the effect on breastfeeding based on lingual frenulum attachment.

There is still no established pattern for the electrical activity of the suprahyoid muscles involved in the sucking function in infants based on lingual frenulum attachment to the tongue and the floor of the mouth. In this context, we note the importance of in-depth studies on this subject that may favor the understanding of muscle activity and aid in the identification of possible oral motor dysfunctions and feeding efficiency problems. These studies may also contribute to the planning and implementation of actions in health services that would enable breastfeeding and minimize the impact and consequences related to early weaning, which would benefit child development.

The aim of the present study is to analyze the electrical activity of the suprahyoid muscles in infants, based on the lingual frenulum attachment to the tongue and the floor of the mouth, during the sucking function in breastfeeding.

## 2. Materials and Methods

### 2.1. Study Design

This is an observational, analytical, cross-sectional study. The Human Research Ethics Committee of the Federal University of Goiás approved the study under number 705.229 on 30 June 2014. This study was conducted at a university in the central region of Brazil. All adults responsible for the infants participating in this study received and signed an informed consent form and a consent form for the participation of a person as a subject, respecting resolution 466/2012

### 2.2. Participants

Participants included full-term infants of both genders, aged between 1 and 4 months old, who were clinically stable and had a birth weight of 2500 to 4000 g, who attended Hearing Health Reference Center at PUC Goiás/Brazil, and were referred for evaluation of the lingual frenulum during the period of March 2015 to December 2016. Exclusion criteria were infants with anatomical and physiological changes on the face, pre-term and post-term neurological impairment, a weight below 2500 g, or those exclusively bottle-fed. 

Sample selection was non-random and performed via quotas/segmentation. The sample was calculated based on the quantity of live births from the Goiânia, Goiás, Brazil. Thus, it took at least 207 participants to reject the null hypothesis and to achieve a 95% test power and type I error of 5% possibility.

### 2.3. Procedures

The suprahyoid musculature (digastric, mylohyoid, geniohyoid, and stylohyoid) was defined for analysis in this study (Figure 1), due to its direct participation in the suction function [17], in relation to the movement and stabilization of the mandible and in the movement of the tongue, as well as its location, which makes it possible to fix electrodes, as compared with other facial muscles [7].

Two phonoaudiologists and seven collaborators (volunteer students) participated in the evaluation of the lingual and electromyographic frenulum, also helping to organize the collected data.

All volunteer students were equally trained, which means that they received previous training to carry out the evaluations with the infants. This training lasted for three months and was carried out by two researching speech therapists. The meetings were weekly, lasting two hours. It involved the study of scientific articles on the topic, the performance of exams supervised by speech therapists and the study of cases.

We used the “protocol for the evaluation of the lingual frenulum in infants” proposed by Martinelli [27].

The parents were advised at the time of the examination not to feed the babies for at least two hours before the evaluation.

In one room, a phonoaudiologists and four collaborators (volunteer students) performed the first and second stage of the protocol.

The first stage of the protocol was the data collection of the clinical history and the second stage of the protocol included an anatomical and functional evaluation.

During the anatomical and functional evaluation, we observed the posture of the lips at rest (closed, semi-open, or open), the tendency of tongue positioning during crying (raised, midline, midline with lateral elevation, or downward tongue tip with lateral elevation), and the shape of the tip of the tongue when raised during crying or a lifting maneuver (rounded, slight crevice at apex, or heart-shaped). Through the elevation of the lateral margins of the tongue by the gloved right and left index fingers of the evaluator, it was observed whether the lingual frenulum could be visualized or not and if it was necessary to be visualized with a specific maneuver. If the lingual frenulum could be visualized, we determined whether it was thin or thick, whether the lingual frenulum was attached to the middle third of the tongue, between the middle third and the apex, or to the apex, and whether the attachment to the floor of the mouth was visible from the sublingual caruncles (opening of the ducts of the right and left submandibular glands) or from the lower alveolar ridge. 

In another room, the other phonoaudiologists and three collaborators (volunteer students), carried out the assessment of the non-nutritive sucking and of the activity of the suprahyoid muscles during nutritive sucking on mother’s breast, using surface electromyography.

Non-nutritive sucking was evaluated by introducing the gloved little finger into the baby’s mouth during sucking for 2 min, and tongue movement was observed (here, adequate was defined as tongue anterioration, coordinated movements, and efficient suction, and inadequate was defined as limited tongue anterioration, uncoordinated movements, and a delayed onset of suction). 

The evaluation of nutritive sucking was made along with surface electromyography. We observed the sucking rhythm (several sucks followed by short pauses, or a few sucks followed by long pauses), coordination between sucking/swallowing/breathing, which was classified as either adequate or inadequate, whether the infant bit the nipple, and whether the infant had tongue snapping during nutritional sucking.

For evaluation of the electrical activity of the suprahyoid muscles (MEA), we used the MIOTOOL 200 device (manufactured by Miotec Equipamentos Biomédicos Ltd, Porto Alegre, Brazil) that comprised four channels, with a data acquisition system that allowed for the selection of 8 independent gains per channel. We used a gain of 1000, two SDS500 sensors (MIOTEC^®^, Porto Alegre, Brazil), a reference cable (earth), and a calibrator (MIOTEC^®^, Porto Alegre, Brazil). We used a 7.2 V 1.700 Ma NiMH rechargeable battery (MIOTEC^®^, Porto Alegre, Brazil), which was operated in isolation from the utility system, and was connected to a Sony Vaio^®^ notebook (Japan Industrial Partners, Nagano, Japan), equipped with an analog-to-digital converter and the Miograph 2.0 software (MIOTEC^®^, Porto Alegre, Brazil).

The records were made in a quiet place with natural light and ambient temperature [28]. We used disposable unipolar surface electrodes (Meditrace^®^, Infant Model, manufactured by Tyco/Kendall-USA, imported by Lamedid Comercial e Serviços Ltd.a, Barueri-SP-Brazil, São Paulo, Brazil). These were made from silver-silver chloride (Ag-AgCl), adhesive, and a conductive solid gel (hydrogel) that was responsible for capturing and conducting the signal of the EMG. 

Subsequently, the skin was cleaned with gauze soaked in 70% alcohol to remove oil or any material would interfere with the capturing of signals [29,30].

The positioning of the three electrodes followed a standard procedure. This started with the reference electrode or “earth”, which was placed on the frontal bone (forehead), minimizing interference from external electrical noise [29]. This was followed by the attachment of the others two electrodes to the suprahyoid muscles, which were placed in a bipolar configuration, with a minimum distance of 10 mm between them [30]. The evaluator stimulated non-nutritive sucking by introducing a flavored (milk breast) finger for five seconds to locate this region and, by this maneuver, it was possible to palpate the muscles.

After attaching the electrodes to the skin of the infant, the clamp sensors were placed following the same order as the electrode attachment [29]. After completing this procedure, the configuration, channel enablement in the software, and subsequent calibration were performed. The three unused channels were disabled.

Subsequently, the electrical activity of the suprahyoid muscles during breastfeeding was recorded for 3 min (Figure 2). The recording was interrupted whenever the signals presented interference noise or were low or absent, where the position of the electrodes was corrected and the exam was started again. The mothers could stop the evaluation at any time and withdraw from the study without penalty.

The mothers received instructions about their positioning, namely, to be seated with their feet on the ground, and about the positioning of the infant, which was to be supported and have the head and the spine aligned straight, with their belly facing the body of the mother and face towards the mother’s breast. The infant’s mouth was facing the areola and nipple in order to catch the breast. Before the evaluation, the mother received the necessary guidelines on the clinical examination, evaluation of nutritive sucking, and electromyographic examination.

The infants whose results showed interference of the lingual frenulum in the tongue movements and/or those with reduced activity of the suprahyoid muscles were referred to basic health units with a speech-language pathology report of the lingual frenulum. The infant was referred to the pediatric dentistry service for assessment and the definition of future conduct.

The data collected in this study were archived in a database that was created using Microsoft Excel^®^ (Microsoft, Redmond, WA, USA) to organize them.

### 2.4. Electromyographic Analysis

The analysis of the electromyographic data was performed by phonoaudiologists professional with experience in this area, who participated in the study specifically for this data analysis.

The Miograph 2.0 (MIOTEC^®^, Porto Alegre, Brazil) software was used for presenting and interpreting the electromyographic signal. The numerical data were expressed in the root mean square (RMS) form, which represents the square root result of the mean square of the instantaneous amplitudes of the signal of the recorded electromyographic trace, the unit of which is expressed in microvolts (μV).

In order to select the best signals, a low pass filter of 20 Hz and a high pass filter of 500 Hz was used. The best configurations that presented the least noise and the most symmetrical and connected histogram to the signal were considered.

The means values, recorded in μV, were transformed into percent values of the reference value (normalization by the maximum peak) for each subject. The formula for calculating the percentage, according to the recommendations of the International Society of Electrophysiology and Kinesiology (ISEK) [30], was (X/Y) × 100, where X is equal to the muscular electrical activity (MEA) mean in the requested task (μV) and Y is equal to the reference value, corresponding to the mean of the MEA in peak (μV). Thus, the highest value of the electromyographic signal of the suprahyoid muscles was identified for 3 s of breastfeeding. The maximum peak was considered to be 100% of activity and the mean activity during 3 s of the breastfeeding was considered as “X” [28,31].

The normalization technique is essential so that the surface electromyography signal can be compared in different studies, muscles, and participants after being analyzed. This technique is considered a prerequisite for any comparative analysis of EMG signals [28,31].

### 2.5. Statistical Analysis

The data were entered into IBM SPSS Statistics version 23 (IBM, Inc., Chicago, IL, USA) and subjected to a descriptive, inferential, and analytical statistical analysis using the frequencies, central tendencies, and dispersion measures. A test power of 95% was used, and the type 1 error was set at 5%.

The muscular electrical activity (MEA) was found to correspond to the dependent variable, which had a normal distribution (Kolmogorov–Smirnov *p* value of 0.113) when considering both groups. The independent variables were the age in days at the electromyographic examination, gender, feeding type, lingual frenulum attachment type, and nutritive sucking pattern. 

The association between the dependent variable RMS and the independent variable age was tested by a one-factor analysis of variance (ANOVA), which allows for the comparison of three or more continuous variables at a single time, determining whether there is a statistically significant difference.

The association between the dependent variable MEA and the other independent variables was tested by Student’s t-test, which allows for the comparison of two samples using their means for statistical inference in order to verify if there is a statistically significant difference.

The EMG signal of the suprahyoid muscles (MEA) of infants with a lingual frenulum without and with ankyloglossia, both as obtained in a crude form (RAW) and rectified (RMS), are shown in Figure 3a,b.

## 3. Results

### 3.1. Characteristics Data of the Participating Infants 

Two hundred and fifty-one infants who met the inclusion criteria were invited and all mothers agreed to participate in the research. After agreement, the infants underwent surface electromyography. Eleven infants were excluded because their records showed interference in the electromyographic signal, and five infants were excluded due to a failure in the capture and recording of the electromyographic signal at the time of evaluation, that resulted in a final sample of two hundred and thirty-five infants. 

Characteristics of the data of the participating infants at the time of evaluation, age at the examination day, sex, and type of feeding are described in Table 1.

In the present study, there was no significant difference between the electrical activity of the suprahyoid muscles during nutritional sucking and the infants’ age in days (*p* = 0.368), gender (*p* = 0.136), and feeding type (*p* = 0.689). 

### 3.2. The location of the Frenulum Attachment

Concerning the location of the frenulum attachment on the tongue and floor of the mouth, 53 (22.5%) infants had the lingual frenulum attached of the middle third on the tongue, visible from the sublingual caruncles, 35 (14.8%) had the lingual frenulum attached to the middle third/lower alveolar ridge, 28 (11.9%) were attached between the middle third and apex, visible from the sublingual caruncles, 102 (43.4%) had the frenulum attached between the middle third and the apex, visible from the lower alveolar ridge, and 17 (7.2%) had the frenulum attached to the apex, visible from the lower alveolar ridge. No infants with the lingual frenulum attached in the apex on the tongue, visible from the caruncles, were identified.

### 3.3. Electrical Activity According to Location of Frenulum Attachment on the Tongue and Floor of the Mouth

We observed (Table 2) a greater electrical activity in the muscle during breastfeeding in the infants with their lingual frenulum attached to the middle third and visible from the sublingual caruncle (40.7%). We found lower electrical activity in the muscle in the infants with their lingual frenulum attached to the apex of the tongue and visible from the lower alveolar ridge (29.9%). 

### 3.4. Electrical Activity of the Suprahyoid Muscles of Infants during Breastfeeding According to Thickness, Frenulum Attachment to the Tongue and Floor of the Mouth, and Nutritive Sucking Pattern

There was a statistically significant difference with a higher mean percentage of electrical activity of the suprahyoid muscle during breastfeeding in infants with the lingual frenulum attached to the middle third/sublingual caruncles, and those who had a thin lingual frenulum (*p* = 0.002), with several sucks and short pauses during nutritive sucking (*p* = 0.003), coordination between sucking, swallowing, and breathing function, without signs of stress (*p* = 0.001), the absence of biting (*p* = 0.005), and the absence of tongue snapping (*p* = 0.001) when compared to infants with the lingual frenulum attached to the middle third/lower alveolar ridge. A higher mean electrical activity of the suprahyoid muscles was observed during nutritive sucking in infants with the lingual frenulum attached to the middle third/sublingual caruncle, who had a thin lingual frenulum (*p* = 0.001), with coordination between sucking (*p* = 0.003), swallowing, and breathing functions without signs of stress (*p* = 0.001), the absence of biting (*p* = 0.001), and the absence of tongue snapping (*p* = 0.001) when compared to infants with the lingual frenulum attached to the middle third and apex/lower alveolar ridge. A statistically significant difference was observed with a higher mean percentage of the electrical activity of the suprahyoid muscles during breastfeeding in infants with the lingual frenulum attached to the middle third/sublingual caruncles, who had a thin lingual frenulum (*p* = 0.02), a NS pattern with coordination between sucking, swallowing, and breathing function without signs of stress (*p* = 0.005), the absence of biting (*p* = 0.03), and the absence of tongue snapping (*p* = 0.006) when compared to infants with the lingual frenulum attached to the apex/lower alveolar ridge (Table 3).

## 4. Discussion

There was a higher frequency (43.4%) of lingual frenulum attachment to the tongue between the middle third and the apex as visible from the lower alveolar ridge. These results agree with the literature, where a retrospective study that analyzed 165 protocols for assessing the lingual frenulum of full-term babies aged between 1 and 4 months old found a prevalence of 32.2% for the lingual frenulum being attached between the middle third and the apex/lower alveolar ridge [32]. Another previous longitudinal study, which evaluated the anatomical characteristics of the lingual frenulum of 71 full-term infants, found that 27 (38%) had their lingual frenulum attached between the middle third and the apex at the 1st, 6th, and 12th months of life, and 42 (59.1%) infants had their lingual frenulum attached in the lower alveolar ridge [15].

In the present study, we found the lower electrical activity of suprahyoid muscles during the sucking of breasts in infants with their frenulum attached of the middle third of the tongue, between the middle third and the apex, and to the apex, as visible from the lower alveolar ridge, regardless of age. These results indicate that the attachment of the lingual frenulum on the mouth floor, as visible from the lower alveolar ridge, appears to interfere more with the forward part of the tongue when compared to the frenulum attached of the middle third of the tongue and the middle third apex.

These results provide important data for the differential diagnosis of the lingual frenulum, because they identify anatomical characteristics that may reduce muscle activity due to the restriction of tongue-tip movement during sucking in infants, which could consequently impair breastfeeding. 

We also observed greater electric activity of the muscle in infants with a thin lingual frenulum and with the frenulum attachment to the middle third of the tongue, as visible from the sublingual caruncles, which had a proper sucking rhythm. A statistically significant difference was found when compared to the muscular electrical activity of infants with the frenulum attached of the middle third of the tongue, between the middle third and the apex, and to the apex visible from the lower alveolar ridge, which had an inadequate sucking rhythm.

A previous study [15] that analyzed 71 full-term infants showed a statistically significant relationship between the anatomical characteristics of the lingual frenulum and nutritive sucking, with the improvement of the sucking pattern in breastfeeding after a frenotomy. Another retrospective study [33] identified that infants with an altered frenulum were more likely to have difficulty sucking.

Previous studies have shown that the tongue actively participates in sucking and is essential for the proper removal of breast milk [8,9,10,11]. The nipple is compressed from the tip to the base when the tongue is high, and, when it lowers, the nipple expands by approaching the hard palate with the soft palate and increases in diameter, causing a vacuum and allowing the milk to flow into the intraoral space [8,9,10,11]. Alterations of the lingual frenulum may limit its mobility [8,12,13,14,15,33,34], resulting in inadequate catching and potentially leading to changes in sucking function, especially in the dynamic of sucking/removal of milk. The main problems identified in cases of an altered lingual frenulum in relation to breastfeeding in the mother’s womb are difficulties in catching, nipple pain and cracking, prolonged feeding times, reduced milk intake by the infant, the loss of weight [12,14,34,35,36,37], dehydration, and growth deficiency [9,32,33,34]. Such changes may hinder the continuity of breastfeeding [12,14,35,36,37], with consequences for the infant’s health, and, later on, the development of chewing and speech [13,15]. 

No studies associating the electrical activity of the suprahyoid muscles and the lingual frenulum of infants during feeding were found in literature. A previous study [7,17,20,22,24] that analyzed suprahyoid muscles during feeding using surface electromyography differs from the present study in terms of the objectives, infant age, sample size, and feeding types, and, because they did not associate the anatomical characteristics of the lingual frenulum, they made it impossible to compare the findings. 

Surface electromyography has been shown to be effective in the early diagnosis of the limitations of tongue movements caused by the lingual frenulum, whose attachment location, both on the tongue and the floor of the mouth, may raise doubts about the reduction of tongue mobility. Thus, it is possible to identify oral motor dysfunctions, enable direct therapeutic interventions and early intervention, and consequently prevent feeding and communication alterations. 

Among the main limitations of our study are the reduced schedules for the examination of the infants due to technical problems related to the regulation of the patients by the Health Unic System, as well as the non-attendance of scheduled infants due to health problems and a public transportation strike.

Further studies investigating the electrical activity of the suprahyoid muscles through surface electromyography after surgery to release the lingual frenulum may contribute to a better understanding of the impact of ankyloglossia on tongue movement during breastfeeding.

## 5. Conclusions

Greater electrical activity of the muscle was observed during the sucking of the maternal breast in full-term infants when the lingual frenulum was attached of the middle third of the tongue and was visible from the sublingual caruncle. Lower electrical activity of the suprahyoid muscles during the sucking of the maternal breast was observed in full-term infants with the lingual frenulum attached at the apex of the tongue and visible from the lower alveolar ridge. 

We also observed greater electrical activity of the muscle in infants with a thin lingual frenulum and a rhythm comprising several sucks and short pauses, an adequate and balanced coordination between the feeding efficiency, and sucking, swallowing, and breathing functions without signs of stress, and absence of biting of the nipple and tongue snapping. 

## Figures and Tables

**Figure 1 ijerph-17-00859-f001:**
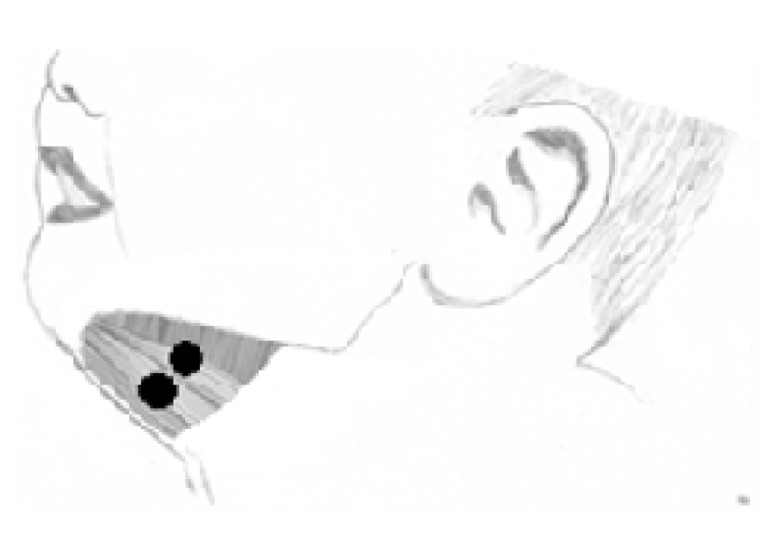
Suprahyoid muscles. Schematic illustration of the muscles analyzed in this study and the location of the surface electromyography electrodes.

**Figure 2 ijerph-17-00859-f002:**
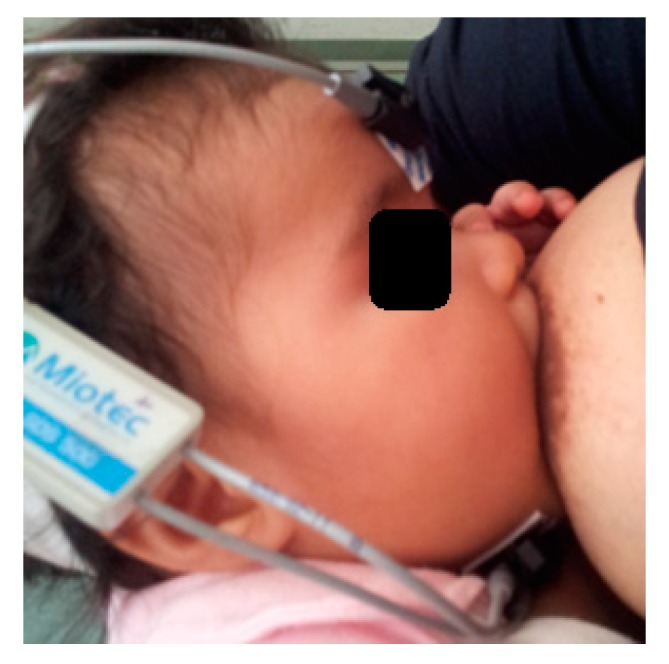
Electromyographic evaluation. Electrodes attached to the bone (forehead) and submandibular (suprahyoid muscles) regions during sucking in breastfeeding.

**Figure 3 ijerph-17-00859-f003:**
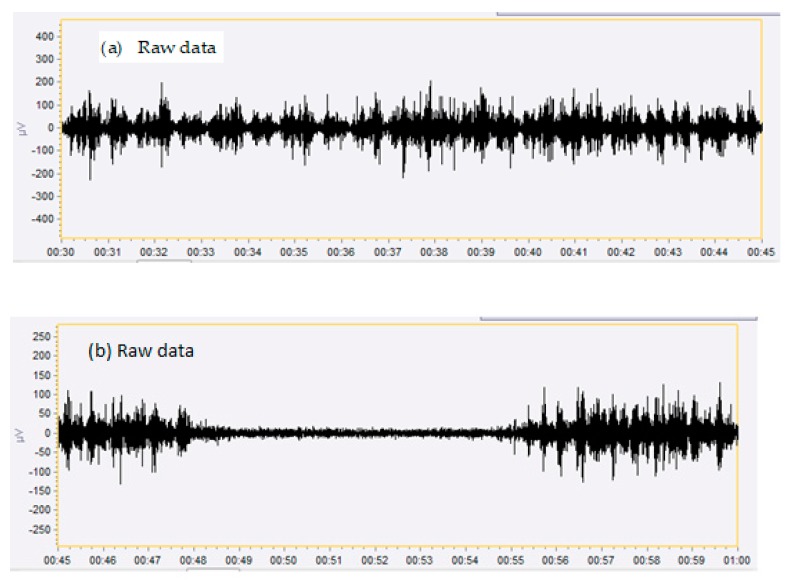
(**a**) A sample of the gross surface electromyography signal register (RAW) of the suprahyoid musculature of infants with lingual frenulum attachment in the middle third and sublingual caruncles and (**b**) lingual frenulum attachment at the apex and lower alveolar ridge.

**Table 1 ijerph-17-00859-t001:** Association between the characteristics of infants (gender, feeding type, and age at the examination) and muscular electrical activity (MEA).

Variables		*n* = 235		
		*n*	MEA(%)	*p*
Gender	Female	122	32.8	0.136
	Male	113	35.7	
Feeding Type	Exclusive breastfeeding	176	34.0	0.689
	Non-exclusive breastfeeding	59	34.9	
Age at examination	31–60 days	127	35.6	0.368
	61–90 days	63	33.8	
	91–120 days	33	30.4	
	121–149 days	12	33.5	

MEA: Muscular electrical activity (normalized).

**Table 2 ijerph-17-00859-t002:** Mean MEA of the suprahyoid muscles of infants based on the lingual frenulum attachment to the tongue and the floor of the mouth during breastfeeding.

Location of Lingual Frenulum Attachment		*n*	MEA (%)	SD
Sublingual	Floor of the mouth	53	40.7	15.2
Middle third	Sublingual caruncles	35	31.1	11.7
Middle third	Lower alveolar ridge	28	36.9	12.9
Middle third/apex	Sublingual caruncles	102	31.8	13.8
Middle third/apex	Lower alveolar ridge	17	29.9	19.5
Apex	Lower alveolar ridge	35	31.1	11.7

SD: Standard deviation. MEA: Muscular electrical activity (normalized).

**Table 3 ijerph-17-00859-t003:** Comparison of the MEA of the suprahyoid muscles of infants during breastfeeding according to thickness, frenulum attachment to the tongue and floor of the mouth (MT/SC and MT/LAR; MT/SC and MTA/LAR; MT/SC and A/LAR), and nutritive sucking pattern.

Variables		A	*n*	MEA (%)	*p*	A	*n*	MEA (%)	*p*	A	*n*	MEA (%)	*p*
Frenulum Thickness	Thin	MT / SC	47	41.1	0.002	MT / SC	47	41.1	0.001	MT / SC	47	41.1	0.02
		MT / IAR	35	31.1		MTA / IAR	94	31.6		A/LAR	17	29.9	
	Thick	MT / SC	6	37.0	0	MT / SC	6	37.0	0.8	MT / SC	6	37.1	0
		MT / IAR	0	0.0		MTA / IAR	8	35.0		A/LAR	0	0.0	
Rhythm	SS/SB	MT / SC	46	41.7	0.003	MT / SC	46	41.7	0.003	MT / SC	46	41.7	0.08
		MT / IAR	32	31.7		MTA / IAR	66	33.2		A/LAR	10	31.2	
	FS/LB	MT / SC	7	33.9	0.31	MT / SC	7	33.9	0.4	MT / SC	7	33.9	0.44
		MT / IAR	3	25.2		MTA / IAR	36	29.4		A/LAR	7	28.2	
Coordina tion	P	MT / SC	51	41.4	0.001	MT / SC	51	41.4	0.001	MT / SC	51	41.4	0.005
		MT / IAR	34	31.1		MTA / IAR	91	32.7		A/LAR	12	27.6	
	I	MT / SC	2	23.1	0.61	MT / SC	2	23.1	0.86	MT / SC	2	23.1	0.62
		MT / IAR	1	31.0		MTA / IAR	11	24.7		A/LAR	5	35.5	
Bite the nipple	N	MT / SC	48	41.8	0.005	MT / SC	48	41.8	0.001	MT / SC	48	41.8	0.03
		MT / IAR	26	31.8		MTA / IAR	78	32.3		A/LAR	13	30.3	
	Y	MT / SC	5	30.3	0.84	MT / SC	5	30.3	0.98	MT / SC	5	30.3	0.79
		MT / IAR	9	29.0		MTA / IAR	24	30.2		A/LAR	4	28.8	
Tongue snapping	N	MT / SC	36	42.5	0.001	MT / SC	36	42.5	0.001	MT / SC	36	42.5	0.006
		MT / IAR	18	27.5		MTA / IAR	34	29.3		A/LAR	6	24.6	
	Y	MT / SC	17	36.7	0.71	MT / SC	17	36.7	0.34	MT / SC	17	36.7	0.6
		MT / IAR	17	35.0		MTA / IAR	68	33.1		A/LAR	11	32.8	

Abbreviations: SS/SB—several sucks/short breaks; FS/LB—few sucks/long breaks; P—proper; I—inadequate; N—not; Y—yes; MEA—muscular electrical activity (normalized); A—attachment; MT/SC—middle third/sublingual caruncles (*n* = 53); MT/LAR—middle third/lower alveolar ridge (*n* = 35); MTA/LAR—middle third and apex/lower alveolar ridge (*n* = 102); A/LAR—apex/lower alveolar ridge (*n* = 17).

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
