# Peer review of "Surface Electromyographic Analysis of the Suprahyoid Muscles in Infants Based on Lingual Frenulum Attachment during Breastfeeding"

_ijerph, 2020, doi:10.3390/ijerph17030859_

Round 1

Reviewer 1 Report

The manuscript Electromyographic analysis of the suprahyoid muscles in infants based on the lingual frenulum attachment during breastfeeding correctly identify the gap.

However, I recommend authors to make some changes on the structure of the manuscript.

Firstly, in the introduction, they should increase the number of recent references. And in page 4, line 67-68, they must revise the reference citations in the text 14-13.

Secondly, it is mandatory to improve section 2.2 “Participants”. The authors should describe more specifically the sample.

Thirdly, in Results section, all numerical data must be described with a decimal and separated by a period. Also check this issue in all Tables.

The manuscript could be accepted after minor revision.

Author Response

Reviewer 1

We appreciate the referees’ comments to our manuscript. Their remarks and suggestions were very relevant, and we attempted to meet them all. As there was a need to add references, we also updated the number of citations.

We are looking forward to hearing from you at your earliest convenience.

Sincerely,

Ellia C. L. França

Response to Reviewer 1 Comments

I recommend authors to make some changes on the structure of the manuscript:

Point 1: Firstly, in the introduction, they should increase the number of recent references. And in page 4, line 67-68, they must revise the reference citations in the text 14-13.

Response 1: We agree with your point and we are very grateful for the suggestion to improve the introduction of this study. Encouraged by your comments, we have added new papers that were also included in the reference list, as highlighted in the revised manuscript file: Lee; Binns (2020), Richard et al. (2018), Raheem et al. (2017), and Pires et al. (2012). We have revised the reference citations in the text.

Point 2: Secondly, it is mandatory to improve section 2.2 “Participants”. The authors should describe more specifically the sample.

Response 2: We added the required information (highlighted in manuscript): was described more specifically the sample.

Point 3: Thirdly, in Results section, all numerical data must be described with a decimal and separated by a period. Also check this issue in all Tables.

Response 3: Changes were provided in the Results section. We described all numerical data with a decimal separated by a period, and we have checked all tables.

Reviewer 2 Report

I have carefully read the paper on electromyographic analysis of the suprahyoid muscles in infants based on the lingual frenulum attachment during breastfeeding

Electromyographic provides information on muscle electric activity, not similar to strength, and what the authors describe are association, not necessary reflecting causality, although its reasonable to assume that the ‘extent of lingual frenulum attachment’ (recommend to us tongue tie in the eg key words, as wording as this is much more commonly used in the literature) is reflected in the muscular activity during breastfeeding. Highly recommend to add some 'relevance' in the introduction on tongue tie aspects. 

The hypothesis is plausible in my assessment (link between emg and frenulum position).

Where the investigators blinded for ‘group’ allocation (‘anatomic assessment’ separated and blinded from the EMG signal ?

It is fair to make is more explicit that this is a surface emg, and not a needle emg (title, abstract) ?

Reference values ?

Can the authors elaborate on gender or age-related differences in EMG patterns ?

Methods

What do the authors mean with ‘proper training’ for the EMG application.

Have the authors data on the inter- and intra-rater variability on the Martinelli protocol within their group, and related to this, were assessors blinded for subsequent EMG signal (potential bias)

Minimum finger = ? little finger ? or fifth finger ?

I disagree on the statement that the data will be destroyed. This is for sure against the trend to provide access to the data. In fact, the authors should be able to provide these data in a supplement as excel or similar source document to this paper ?

The abstract suggest that emg signal were compared to reference data, but I could not retrieve a sufficiently clear description on this aspect of the study in the methods section, as it reads that the patient self is the reference.

Statistics: have the authors also considered to perform a multiple regression model analysis or covariate analysis to explore the significance of age, gender and lingual frenulum on the variability in activity observed. In essence, how ‘relevant’ is lingual frenulum anatomy compared to the other covariates. The population is quite big, but has a power calculation been performed before recruitment ?

Results

How many cases were considered, invited versus recruited ? (cfr last alineas of the discussion on recruitment problems)

Author Response

Reviewer 2

We appreciate the referees’ comments to our manuscript. Their remarks and suggestions were very relevant, and we attempted to meet them all. As there was a need to add references, we also updated the number of citations.

We are looking forward to hearing from you at your earliest convenience.

Sincerely,

Ellia C. L. França

Response to Reviewer 2 Comments

Point 1:  Electromyographic provides information on muscle electric activity, not similar to strength, and what the authors describe are association, not necessary reflecting causality, although its reasonable to assume that the ‘extent of lingual frenulum attachment’ (recommend to us tongue tie in the eg key words, as wording as this is much more commonly used in the literature) is reflected in the muscular activity during breastfeeding. Highly recommend to add some 'relevance' in the introduction on tongue tie aspects.

Response 1: We agree with your point and we are very grateful for the suggestion. We have added tongue tie among the key words, and highlighted in the manuscript, in the relevance in the introduction,  about how oral dysfunctions caused by an altered lingual frenulum (tongue tie) may compromise breastfeeding by causing discomfort and pain to mothers, ineffective emptying of the breast, poor weight gain, and/or early weaning.

Point 2: The hypothesis is plausible in my assessment (link between emg and frenulum position).

Response 2: No changes needed.

Point 3: Where the investigators blinded for ‘group’ allocation (‘anatomic assessment’ separated and blinded from the EMG signal?

Response 3: The groups have been defined after anatomical and functional evaluation of the lingual frenulun, according to the determined classification: whether the lingual frenulum was attached to the middle third of the tongue, between the middle third and the apex, or to the apex; and whether the attachment to the floor of the mouth was visible from the sublingual caruncles (opening of the ducts of the right and left submandibular glands) or from the lower alveolar ridge. Subsequently, all infants had the nutritive sucking evaluated during breastfeeding using surface electromyography.

Point 4: It is fair to make is more explicit that this is a surface emg, and not a needle emg (title, abstract)?

Reference values?

Response 4: We accepted your suggestion and added the word ‘surface’ to the title and abstract (highlighted in the revised manuscript).

Point 5: Can the authors elaborate on gender or age-related differences in EMG patterns?

Response 5: Thank you for the suggestion. We agree with you and added the data in the table 1 and expanded interpretation in the last paragraph of the section "Characteristics data of the participating infants ". In the present study, there was no significant difference between the electrical activity of the suprahyoid muscles during nutritional sucking and the infants' age in days, gender and feeding type. 

Point 6: Methods -What do the authors mean with ‘proper training’ for the EMG application.

Response 5: Participated in the anatomical and functional evaluation of the lingual frenulun and electromyographic two professional phonoaudiologists who performed all the evaluations proposed in this study and seven collaborators (volunteer students), properly trained, who assisted in the organization of the data.

Point 7: Methods - Have the authors data on the inter- and intra-rater variability on the Martinelli protocol within their group, and related to this, were assessors blinded for subsequent EMG signal (potential bias)

Response 7: The analysis of all electromyographic data in the infants presented and interpreted the Miograph 2.0 (MIOTEC ®, São Paulo, Brazil) software, and was performed by the speech therapist after the anatomical and functional evaluation of the lingual frenulun and of nutritive sucking along with surface electromyography.

At the time of the analysis, only the infants' initials were recorded, without recording the data concerning the frenulum of the tongue, stored on the Martinelli protocol. We can state that the evaluators were blinded for the analisys of the electromyography signal.

Point 8: Methods - Minimum finger = ? little finger ? or fifth finger ?

Response 8:  We accepted your suggestion and added the word ‘little finger’ (highlighted in the revised manuscript).

Point 9: Methods - I disagree on the statement that the data will be destroyed. This is for sure against the trend to provide access to the data. In fact, the authors should be able to provide these data in a supplement as excel or similar source document to this paper ?

Response 9: Thank you for the suggestion. We added the required information (highlighted in manuscript): The data collected in this study were archived in a database created using Microsoft Excel® to organize them.

Point 10: Methods -The abstract suggest that emg signal were compared to reference data, but I could not retrieve a sufficiently clear description on this aspect of the study in the methods section, as it reads that the patient self is the reference.

Response 10: The numerical data of muscle electrical activity was expressed in root mean square (RMS), which represents the square root result of the mean square of the instantaneous amplitudes of the signal of the recorded electromyographic trace whose unit is expressed in microvolts (μV), was normalized, ie, transformed into a percentage value of the reference value for each infant, which is fundamental, so that the surface electromyography signal, after being analyzed, can be compared in different studies, muscles and participants.

EMG measurements followed the recommendations (exam standardization, definition of the musculature, positioning and distance of the electrodes, cleaning, site, environment) rigorously, according to SENIAM- European Recommendations for Surface Electromyography (2013), De Luca et al. (2006), Ball (2010).

Point 11: Methods - Statistics: have the authors also considered to perform a multiple regression model analysis or covariate analysis to explore the significance of age, gender and lingual frenulum on the variability in activity observed. In essence, how ‘relevant’ is lingual frenulum anatomy compared to the other covariates. The population is quite big, but has a power calculation been performed before recruitment ?

Response 11: We added the required information (highlighted in manuscript): Were included participants full-term infants of both genders aged between 1 and 4 months, clinically stable and birth weight of 2500 to 4000 g, who attended the CRESA - PUC Goiás/Brazil, referred for evaluation of the lingual frenulum, from March 2015 to December 2016.

Exclusion criteria were infants with anatomical and physiological changes on the face, pre-term and post-term neurological impairment, weight below 2500 g or exclusively bottle-fed.

Sample selection was non-random by quotas / segmentation. The sample was calculated based on the quantity of live births in the metropolitan area of Goiânia - Goiás - Brazil. Thus, it took at least 207 participants to reject the null hypothesis and to achieve a 95% test power and type I error of 5% possibility.

                                                                                                                     Point 12: Results - How many cases were considered, invited versus recruited ? (cfr last alineas of the discussion on recruitment problems)

Response 12: We added the required information (highlighted in manuscript): Were included participants full-term infants of both genders aged between 1 and 4 months, clinically stable and birth weight of 2500 to 4000 g, who attended the CRESA - PUC Goiás/Brazil, referred for evaluation of the lingual frenulum, from March 2015 to December 2016.

Exclusion criteria were infants with anatomical and physiological changes on the face, pre-term and post-term neurological impairment, weight below 2500 g or exclusively bottle-fed.

Sample selection was non-random by quotas / segmentation. The sample was calculated based on the quantity of live births in the metropolitan area of Goiânia - Goiás - Brazil. Thus, it took at least 207 participants to reject the null hypothesis and to achieve a 95% test power and type I error of 5% possibility.

Round 2

Reviewer 2 Report

some suggestions at first revision have been bypassed, 

i have also read that all were properly trained, but my question was: what do you mean with this. 

blinding: were those who collected emg data and analysed the data blinded for the 'clinical' frenulum assessment 

finally, 235 cases were included, but how many were invited, how many refused, how many failed ? can you add this info to the paper 

Author Response

Response to Reviewer 2 Comments (2)

Point 1:  Some suggestions at first revision have been bypassed.

Response 1: Thank you for the suggestion. We agree with you that some suggestions at first revision were not enlightened.

Point 2: I have also read that all were properly trained, but my question was: what do you mean with this.

Response 2: We have added the required information (highlighted in manuscript):

We said that they were properly trained because to participate in the research all were equally trained, which means that they received previous training to carry out the evaluations with the infants.

This training lasted for three months and was carried out by two researching speech therapists. The meetings were weekly, lasting two hours. It involved the study of scientific articles on the topic, the performance of exams with infants, supervised by speech therapists and the study of cases that were in phonoaudiology care.

Point 3: Blinding: were those who collected emg data and analysed the data blinded for the 'clinical' frenulum assessment.

Response 3: We have added the required information (highlighted in manuscript):

In one room, a phonoaudiologists and four collaborators (volunteer students) performed the evaluation of the lingual frenulum using the Martinelli protocol.

In another room, the other phonoaudiologists and three collaborators (volunteer students), carried out the assessment of the activity of the suprahyoid muscles during nutritive sucking on mother's breast, using surface electromyography.

The analysis of the electromyographic data was performed by another phonoaudiologists professional with experience in this area, who participated in the study specifically for this data analysis.

Point 4: Finally, 235 cases were included, but how many were invited, how many refused, how many failed ? can you add this info to the paper?

Response 4: Yes, we can add this info to the manuscript. We have added the required information (highlighted in manuscript):

Two hundred and fifty-one infants who met the inclusion criteria were invited and all mothers agreed to participate in the research. After agreement, the infants underwent surface electromyography between February 2015 to November 2016. Eleven infants were excluded because their records showed interference in the electromyographic signal, and five infants were excluded due to a failure in the capture and recording of the electromyographic signal at the time of evaluation, that resulted in a final sample of 235 infants.
